# Atypical Response to Affective Touch in Children with Autism: Multi-Parametric Exploration of the Autonomic System

**DOI:** 10.3390/jcm11237146

**Published:** 2022-12-01

**Authors:** Maria Rosa Bufo, Marco Guidotti, Yassine Mofid, Joëlle Malvy, Frédérique Bonnet-Brilhault, Nadia Aguillon-Hernandez, Claire Wardak

**Affiliations:** 1UMR 1253, iBrain, Université de Tours, Inserm, 37000 Tours, France; 2Centre Universitaire de Pédopsychiatrie, CHRU de Tours, 37000 Tours, France; 3Centre Hospitalier du Chinonais, 37500 Saint-Benoît-la-Forêt, France

**Keywords:** affective touch, discriminative touch, autonomic, pupil, skin conductance, heart rate, Autism Spectrum Disorder

## Abstract

This study aimed at evaluating the autonomic response to pleasant affective touch in children with Autism Spectrum Disorders (ASD) and age-matched typically developing (TD) peers, thanks to multiple autonomic nervous system (ANS) parameters and by contrasting CT (C-tactile fibers) high- vs. low-density territory stimulations. We measured pupil diameter, skin conductance, and heart rate during gentle stroking of two skin territories (CT high- and low-density, respectively, forearm and palm of the hand) in thirty 6–12-year-old TD children and twenty ASD children. TD children showed an increase in pupil diameter and skin conductance associated with a heart rate deceleration in response to tactile stimulations at the two locations. Only the pupil was influenced by the stimulated location, with a later dilation peak following CT low-density territory stimulation. Globally, ASD children exhibited reduced autonomic responses, as well as different ANS baseline values compared to TD children. These atypical ANS responses to pleasant touch in ASD children were not specific to CT-fiber stimulation. Overall, these results point towards both basal autonomic dysregulation and lower tactile autonomic evoked responses in ASD, possibly reflecting lower arousal and related to social disengagement.

## 1. Introduction

### 1.1. Human Processing of Affective and Pleasant Touch

One of the first ways medium of parent–child interaction is touch. This sensory modality emerges very early on, allowing for social communication prior to language development [1,2]. Consequently, touch possesses a social and affective value, crucial in mechanisms such as attachment, bonding and social affiliation [3,4]. Interestingly, affective and social touch seems to be processed by a specific neurophysiological pathway referred to as the “affective touch” pathway and involving specific mechanoreceptors (see [5] for review; [6]). These receptors, the C-LTMRs (C-low-threshold mechanoreceptors), react to gentle stroking of the skin and can be considered caress detectors [7,8,9]. The optimal velocity of the caress to recruit these receptors has been associated with more pleasant judgement [7,9]. The C-LTMRs thus play a fundamental role in the detection and transmission of pleasant stimulations, via C-tactile (CT) fibers, and convey affective tactile information towards the insula [10]. C-LTMRs are almost completely missing from glabrous skin sites, such as the palm of the hand [11]. Glabrous skin is mainly involved in discriminative touch, recruiting A fibers and the primary somatosensory cortex. As a result of this anatomical dissociation, between receptors, peripheral pathways and processing networks, it has been proposed that the CT pathway is the functional channel processing the affective, social and emotional content of a tactile stimulation [5,12].

Nonetheless, pleasant and affective contact can arise from the stimulation of the glabrous part of the hand [13,14], but it is not clear if this perception is mediated by the low density of CT afferents [11]. Indeed, studies in patients show that the section of the CT tract spares pleasant affective touch perception [15], suggesting an involvement of A fibers in this percept, possibly via an activation of insula, cingulate regions and amygdala [16,17]. Mechanisms underlying the perception of the pleasantness and affective value of a tactile interaction are thus not yet completely understood (e.g., [18,19,20]), especially as, in typical individuals, it is impossible to stimulate CT fibers without also stimulating co-localized A fibers. In the present study, we will use the term affective touch for any tactile stimulus corresponding to a caress or a brush of the skin producing a pleasant perception.

Gaining knowledge on touch processing is also crucial as it could play a key role in understanding the sensory and social difficulties encountered in some neurodevelopmental disorders, such as Autism Spectrum Disorders (ASD) [21].

### 1.2. Affective Touch Processing in ASD

ASD is a group of neurodevelopmental disorders characterized by two main features: impairments in communication and deficits in social interactions on one hand, and presence of restrictive repetitive and stereotypical behaviors, interests, or activities on the other hand [22]. Included in these core domains, ASD children and adults often display sensory perceptual anomalies [23], usually described as hypo- or hyper-reactivity to sensory input, that impact ASD individuals’ daily life and societal inclusion [24,25]. Sensory testing has thus become crucial in clinical evaluation, to improve tailored intervention.

Given the importance of affective touch in social relationships, exploring tactile perception in individuals with ASD is of particular interest. Acute tactile sensitivity or the inability to modulate tactile input is hypothesized to impede social behavior that involves interpersonal touch [26]. Behaviorally, most studies investigating tactile dysfunction in ASD have focused on parents’ and teachers’ reports, exploring subjective assessments of both behavioral and emotional responses to touch. These sensory questionnaires, such as the Short Sensory Profile [24,27], show tactile dysfunction in ASD children (e.g., [27,28,29]) compared to typically developing (TD) children, or even to other developmental disorders [30], even if not always replicated [31]. Very few tactile psychophysics studies have been conducted, but they have suggested hypersensitivity to some aspects of tactile stimulations in ASD [32,33], or in some cases reduced tactile discrimination [34], or even no change in tactile discrimination in ASD [31,35,36,37]. However, these psychophysics results may not be representative of the whole autism spectrum, as these studies require active manual or oral participation and lengthy assessments. Participants with more severe symptoms are thus tested less, while tactile atypical responsiveness has been shown to be correlated to the severity of core ASD symptoms [21,35,38].

Focusing on affective touch, hedonic evaluations of textures on the forearm and the palm of the hand show that adults with ASD globally judge the pleasantness similarly to TD adults, but with more variability and extreme scores [33,39]. In children with ASD, contradictory reports have been made, with either lower [40] or higher [41] pleasantness ratings for brushing textures than in TD children. Moreover, affective touch in ASD children evokes defensive reactions more severe in regions with a high density of CT fibers, such as the forearm or the face, than in low-density CT regions, such as the palm of the hand [40], and neither adults nor children with ASD find affective touch reinforcing [42]. Both impaired tactile discrimination and defensiveness to gentle touch are associated with developmental effects on social behavior in ASD mouse models [43] and aversion to social touch has been identified among several atypical behaviors seen in infants later diagnosed with autism [44]. 

Characterizations of neurophysiological and cerebral responses to affective touch in ASD have mainly been done with functional imaging. Several studies compared high-density CT-fiber region (forearm) to low-density CT-fiber region (palm of the hand) stimulation in ASD individuals. In ASD children and adolescents, stimulation of the forearm led to diminished limbic activations correlated with the ASD severity score, while stimulation of the palm of the hand led to a larger activation in somatosensory areas, compared to TD participants [41]. In TD adults, limbic system activation was larger for slow (optimal) brushing of the CT fibers compared to fast (non-optimal) brushing of the CT fibers [45]. More precisely, the activation of the superior temporal sulcus (STS) was inversely correlated to the autistic traits of participant [45], whereas it was positively correlated to the awareness of affective touch in TD, but not in ASD participants [46]. In addition, in adults with ASD, the stimulation of the forearm with a pleasant or neutral texture led to a reduction in somatosensory region activations (compared to TD adults), while an unpleasant texture evoked larger activations in somatosensory and insular regions correlated with the social score of ASD participants [39]. Globally, these studies suggested that pleasant stimulations targeting the CT fibers evoked smaller responses in the limbic system of individuals with ASD. Some EEG studies seem to point in the same direction [47]. However, functional imaging, as with psychophysics studies, possibly bias their sample of ASD participants towards individuals with fewer sensory abnormalities. One of the objectives of the present study was to explore alternative neurophysiological evaluations of tactile stimulations, specifically autonomic indices, which could be more easily applicable to the whole ASD spectrum [48].

### 1.3. Autonomic Reactivity to Affective Touch

Even though affective touch has been explored in humans, thanks to behavior, functional imaging and electroencephalography (e.g., [49,50,51]), its influence on autonomic measures has just started to be explored. The relevance of autonomic responses is that they are objective, automatic and easily accessible in participants, whatever their age or condition (for example in non-verbal patients). Previous results lead us to believe these measurements could be appropriate to study affective touch: first, a simple sensory stimulation has proved to evoke an autonomic response in several sensory modalities (visual, auditory and tactile) [52,53,54] measured with several autonomic indices (e.g., heart rate, skin conductance and pupil diameter); second, the autonomic nervous system (ANS) reacts to emotional situations [55] and affective touch evokes pleasantness or emotional sensations [49]. 

Previous studies have shown that affective tactile stimulations led to an activation of the sympathetic branch of the ANS (sympathetic nervous system, SNS) and also an activation of the PNS (parasympathetic nervous system). The SNS tactile activation has been observed in adults via an increase in the pupil diameter [56,57] (see review [58]) and of the skin-conductance phasic activity [57]. However, these studies did not point towards a SNS activation specific to CT fiber stimulation, with larger activations for faster stimulations [57,59]. The PNS mobilization has been evidenced by a decreased heart rate in 9- month-old infants for CT-optimal stroking [60,61], and is accompanied by beneficial effects [62] on cardiorespiratory stabilization in infants [63]. In adults, Pawling et al. [64,65] reported that heart rate deceleration was larger after CT fiber optimal than CT non-optimal stimulation of the forearm, but with no difference for forearm (high-density CT) or palm of the hand (low-density CT) stimulations.

In line with these overall results, in their recent review Walker et al. [20] proposed that, if the CT fibers “play a specific role in signaling the social affective relevance of touch”, a CT-targeted stimulation should induce an initial PNS orienting response, visible in a decreased heart rate, and an increase in SNS response, visible in a skin conductance and pupil diameter increase. A non-CT-targeted stimulation would also produce a SNS response, but with an amplitude related to the intensity of the stimulation (in velocity or pressure), while the PNS response may depend on the motivational value of the stimulation.

Considering this evidence, we chose to study three autonomic indices simultaneously: the pupil diameter, the skin conductance reactivity (SCR) and the heart rate (HR).

### 1.4. Autonomic Activity in TD and ASD Populations

The ANS is necessary to maintain stable conditions in our organs (homeostasis) and indispensable to survive [66]. The two branches of the ANS, the SNS and the PNS, are involved, respectively, in high- and low-arousal states, preparing the body for strenuous physical activity or resting functions [67]. The two branches, however, are not necessarily antagonistic, with possibilities of coactivation, coinhibition, and uncoupled modes of functioning [68,69]. Extending Cannon’s fight or flight theory [66], Porges’ Polyvagal theory [70,71] focused on the cardiac vagal tone, under PNS control, proposing it could reflect panic [72] but also empathy and engagement responses [73,74,75]. 

The heart is regulated both by a dual vagal PNS innervation (dorsal motor vagal branch and smart vagus, originating in the nucleus ambiguous) [76] and SNS innervation. As a result, HR can increase in fight and flight situations, and decrease in freezing and social engagement situations. The pupil diameter, controlled by the iris muscles, is also under the influence of the PNS (through the Edinger–Westphal nucleus) and the SNS (through the ciliospinal center and superior cervical ganglia) (e.g., [77]). In an environment with stable luminosity, pupil diameter increases following sensory, salient, emotional and cognitive stimulation. The skin conductance, reflecting the opening of the sweat glands, is under SNS influence only [78], and is frequently used to index emotional responses (e.g., [79]). Recording multiple autonomic indices simultaneously can provide access to the global state of mobilization of the whole organism: rest, engaged, flight or fight, or freezing. To our knowledge, only three studies so far have recorded at least three indices simultaneously in TD adults following a sensory stimulation and showed the co-occurrence of PNS and SNS activations: (a) pupil dilation, skin conductance increase and heart rate deceleration in response to emotional pictures [55] or faces [79], and (b) skin conductance increase, heart rate deceleration and increased respiratory sinus arrhythmia in response to touch [80]. However, there is also a strong inter-individual variability in the autonomic physiological response, for example in stressful situations [81,82]. Individual autonomic responses could also be directly correlated to temperament [76,83] or personal history [84].

Individuals with ASD appear to have an atypical ANS functioning, both at rest and in other contexts [85,86,87]. Most studies suggested the presence of a SNS hyper-arousal in ASD (but see [88]), possibly linked to social and emotional difficulties [86]. ANS response to sensory stimulations are less documented. When viewing faces, children with ASD showed lower RSA amplitude and faster HR [89,90], larger SCR [91], and lower pupil dilation [92,93] than TD peers. For tactile stimulations, one study showed that children with ASD exhibited lower baseline skin conductance and decreased SCR compared to TD children [88] following stroking with a feather (without precision of the location of stimulation). On the contrary, Fukuyama et al. [94] showed no difference in skin conductance baseline and larger SCR responses in ASD adults compared to TD adults following an electrical stimulation of the forearm. Autonomic responses to touch, and in particular affective touch, are thus not clear in autism. 

Interestingly, autonomic responses to sensory stimulations in ASD could either complement clinical sensory questionnaires, going further into the physiological understanding of observable sensory atypicalities, but could also reveal unobserved sensory dysfunctioning. One could propose that hyporeactive (respectively, hyper-) ASD participants would exhibit lower (respectively, larger) physiological responses, and thus that ANS measures could correlate with sensory questionnaires scores. No study has directly tested this hypothesis, but two studies suggest that this link may not be direct. First, McIntosh et al. [95] showed that children with sensory-modulation disruptions have an overall larger electrodermal response to sensory stimulations than TD children, but with no difference in sensory profile scores between SCR hypo- and hyper-responders. Secondly, in a context of reactivity to painful tactile stimuli, Tordjman et al. [96] showed that an absence of observable reaction in ASD children could be associated with an increased heart rate, showing a dissociation between physiological response and behavioral response. Autonomic physiological response to sensory stimulation would then be an invaluable help to characterize clinically ASD participants.

Recording simultaneously the pupil diameter, the skin conductance and the heart rate could thus help better understand how affective touch is processed in individuals with ASD.

### 1.5. Objectives and Hypotheses

In this study, we aimed at evaluating autonomic responses to affective touch in children with and without ASD, by simultaneously recording three autonomic indices (pupil dilation, skin conductance and heart rate). We compared identical stimulations (gentle stroking of the skin with a CT-optimal velocity) at two locations: a CT high-density territory (forearm) and a CT low-density territory (palm of the hand).

We had three objectives. The first was to determine whether ANS responses would distinguish CT high- vs. low-density territory stimulations. Our hypothesis was that we would observe in TD children, as in TD adults, a pupil dilation and skin conductance response, indicative of SNS mobilization, and a heart rate decrease, indicative of PNS mobilization, and social orienting following pleasant tactile stimulation [71,97]. Moreover, if CT fibers convey the emotional and affective value of the stimulation, we would expect a larger SNS response following the forearm stimulation [20].

The second objective was to evaluate whether ASD children would exhibit a specific ANS signature compared to TD children, and if so, if this different ANS response would be restricted to the stimulation targeting the CT high-density location. Considering previous studies evaluating ANS reactivity to sensory stimulations in ASD, we would expect both reduced SNS and PNS responses in ASD children compared to TD children. These lower responses would possibly be linked to lower motivation and reduced social affiliative behaviors [71,98,99], and as a consequence, the difference in response between the two locations of stimulation possibly observed in TD children would disappear in ASD children. 

Finally, we wanted to evaluate how our three ANS indices were correlated to each other and to sensory and clinical profiles. Sensory profiles measured by questionnaires show differences between individuals with and without ASD, that could be reflected or not in the ANS measures. These physiological measures could thus be used as a complementary sensory evaluation for clinical profiles. Moreover, several studies have described correlations between autism severity scores and cerebral activation amplitude in response to pleasant touch, suggesting that children with more severe autism would exhibit the largest ANS reactivity differences to TD children.

This study thus aims at deepening our understanding of tactile processing in ASD, to better understand if precise neurophysiological pathways (tactile and autonomic) are specifically affected in ASD, and to evaluate if this neurophysiological sensory evaluation can bring Appendix A to complement classical clinical sensory evaluations.

## 2. Materials and Methods

### 2.1. Participants

Thirty TD children (13 boys; 6–12 years old, mean age 9.2 years ± SD 0.2) were recruited according to the following criteria: aged between 6 and 12, no psychiatric disorders or neurologic diseases previously diagnosed, and no learning disabilities. Absence of medical conditions and disabilities have been confirmed by interviewing parents and by looking at their medical booklet. Thirty-seven children with ASD aged between 6 and 12 years were recruited from the University Hospital of Tours, EXcellence in Autism Center–Tours. The ASD diagnosis was made by a child psychiatrist according to the DSM-5 [22] criteria, complemented by ADI-R [100] and/or ADOS-2 [101] assessments, after an extensive evaluation including detailed developmental history, psychiatric assessment and pediatric, psychological and neurological examinations. Exclusion criteria were the use of medical treatments affecting neural activity. However, only children with ASD who first accepted to sit on the armchair, second tolerated the physiological captors (see Section 2.2), and third stayed calm during the data acquisition were considered for the analyses. The final ASD group was thus composed of twenty children (19 boys; 6–12 years old, mean age 9.1 years ± SD 1.7) while 17 ASD children could not be tested or included in the analyses. The main characteristics of the TD and ASD participants are summarized in Table 1 (see also Appendix A.

The age range (6–12 years old) was chosen according to the clinical population we could access and the fact that these children find gentle stroking stimulation pleasant [102], as in adults.

The children gave verbal consent and their parents provided written informed consent according to institutional guidelines. The experiment conformed to the Code of Ethics of the World Medical Association [103] and was approved by a French ethics committee before it started (PROSCEA CPP 2017-A00756-47 and SCOPA CERNI 2017-12-02). All the participants were recorded in the same conditions, in the University Hospital of Tours, France.

### 2.2. Material

Tactile stimulations were delivered by the experimenter thanks to a manual tool designed and 3D-printed in the laboratory. The extremity of the tool consisted of a pressure captor covered by the tactile texture, a small piece of goat fur. That texture was evaluated as pleasant, when brushed against the skin, by a preliminary panel of adults. The participants were installed in an armchair facing a computer screen. Their left arm was encased in a comfortable orthotic device fixed on the armrest in order to limit the arm movements and facilitate the access to the two body zones of interest: the palm of the hand and the forearm.

The cardiac frequency and skin conductance response (SCR) were recorded by using BIOPAC MP36^®^ (BIOPAC Systems Inc., Goleta, CA, USA), with a constant voltage of 0.5 V. To monitor these two signals during the experiment, we used the AcqKnowledge^®^ 4.1 software. The SCR was recorded using two 8 mm Ag/AgCl cup electrodes (EL258; BIOPAC Systems Inc., Goleta, CA, USA), positioned on the second phalange of the index and medium finger of the right hand, and 0.5%-NaCl electrode paste (GEL101; BIOPAC Systems Inc., Goleta, CA, USA). The frequency of acquisition for the SCR was 1 kHz with a range band between 0 and 5Hz. The electrocardiogram (ECG) was recorded using two disposable vinyl electrodes (EL503; BIOPAC Systems Inc., Goleta, CA, USA) placed on the sternum and on the right shoulder. The frequency of acquisition for the ECG was fixed at 2 kHz (range band 0–35 Hz). The BIOPAC system also allowed us to record the exact timing and pressure of the tactile stimulation thanks to the pressure captor fixed underneath the texture (acquisition frequency 2 kHz). The pupil diameter was measured by using an eye tracking system SMI RED500^®^ (acquisition frequency 500 Hz) synchronized with BIOPAC.

Other assessments were mainly paper and pen questionnaires. Sensory assessment was done by using Short Sensory Profile 2 (SSP2) [104], a standardized questionnaire of sensory modulation function. The questionnaire is composed of 34 items distributed across four sensory modulation quadrants: seeking (7 items), avoiding (9 items), sensitivity (10 items), and registration (8 items). Parents score the frequency of each item on a Likert scale 1 (almost never = 10% or less) to 5 (almost always = 90% or more). Severity scores are divided into five ranges (much less than others = 0–[6–7]; less than others = [7–8]–[17–20]; just like the majority of others = [18–21]–[42–47]; more than the others = [43–48]–[53–60]; much more than others = [54–61]–[95–110]). Intellectual quotients (IQ) were estimated in ASD children with different tests depending on age and ability (Wechsler intelligence scales [105]: WIPPSI-III, WISC-IV; BL-R Brunet-Lézine scale) and we gathered the global IQ composite scores (verbal and non-verbal). TD children were evaluated for four IQ subtests (cube, matrix, vocabulary and similitude) of the WISC-IV [105]. These four subtests allowed an evaluation of the composite verbal and non-verbal scores. The Childhood Autism Rating Scale (CARS) [106] was used to determine the severity of autism symptoms.

### 2.3. Procedure

The measurements and the stimulations were all carried out by one out of two experimenters, always in the same experiment room, whose conditions of luminosity (10 Lux), hygrometry (27% rh) and temperature (23 °C/76.4 °F) were constant. 

First, the participant was asked to sit on an armchair opposite a computer screen (resolution 1920 × 1080 pixels, visual angle around 45 × 25°) on which the eye-tracking illuminator and camera were fixed. Optimal distance and height were set up by using the IView-X 2.8 software (SMI^®^) and, when necessary, a booster seat for children was placed on the armchair. Then the experimenter installed all the captors (SCR, ECG).

The experimental phase started with an eye calibration phase, in which participants were asked to follow a white spot to five points on the screen. A rest period of five minutes was then carried out to allow for the stabilization of the physiological constants. During the experiment, only a central black fixation cross on a grey background appeared on the screen. All participants were asked to stay still and to look at the fixation cross during the whole experiment. When necessary, a pause could be inserted in between trials if the child was tired or restless. A total of twenty trials were completed. Each trial started with a 4 s pre-stimulation period, followed by a 4 s tactile stimulation, and an 8 s post-stimulation period (Figure 1). An inter-trial interval of 12–16 s was inserted in between trials in order to ensure each physiological parameter returned to its baseline (in particular the SCR), while still avoiding a high temporal expectation and anticipation. The tactile stimulations consisted of stroking either the dorsal forearm part (high-density CT-fiber area) or the palm of the left hand (low-density CT-fiber area) (Figure 1). The two sites were stimulated in ten trials each, in a pseudo-random order, with no more than three consecutive stimulations at the same site. The stimulation itself consisted of stroking the tool covered by the texture on a 5–6 cm long portion of skin, up and down the arm, at a speed of about 6 cm/s [7,107]. The trained experimenter performing the stimulations and the participant were separated by a panel, which left space enough to allow the tool to access the arm but occluded the vision of the experimenter in order to avoid an anticipation effect.

During the experiment, but in another room, parents of TD children were asked to fill in the Short Sensory Profile 2 [104]. TD children were evaluated for the four IQ subtests either before or after the tactile experiment. For ASD children, the Short Sensory Profile 2 was filled in by the parents in coordination with the clinical team. Other scores for ASD children, when present, were collected from the clinical team.

### 2.4. Preprocessing of the Signals

The raw signal was preprocessed with MATLAB^®^ (r2016a; MathWorks).

#### 2.4.1. Tactile Stimulation

As the stimulations were applied manually, data recorded by the pressure captor were used to correct for the exact timing of the stimulation and evaluate the pressure applied. For each trial, the beginning of the stimulation was set when the pressure went above 0.1 mV. The real timing of the stimulation was then superimposed on the acquired signal for each physiological parameter.

#### 2.4.2. Pupil Diameter

Pupil data preprocessing was first required to eliminate artifacts, such as blinking and brief signal losses. To do this, a velocity-based algorithm was used [108,109]. Afterwards, the resulting signal was smoothed over using a median filter and a low pass filter with a cutoff frequency of 4 Hz [109,110]. Pupil size variation was calculated, for each participant and each trial, by subtracting the baseline value from the pupil time course. The baseline value consisted of the mean value of the last 250 ms before the beginning of the stimulation. For each participant, we averaged the trials for each location of stimulation then extracted three parameters: the area under the curve (AUC; in mm.s), the maximum amplitude of response (in mm) and the latency of the maximum response (in s) calculated from the 4 s of the stimulation. We also analyzed the mean baseline value (in mm).

#### 2.4.3. Skin Conductance

Preprocessing of the skin conductance responses (SCR) was performed in Ledalab [111], an open-source software for MATLAB (V3.4.9). The data were downsampled to 10 Hz sampling rate resolution and bandpass-filtered with a first-order Butterworth filter and cut-off frequencies of 5 Hz [112]. Artifacts due to noise were corrected by using the spline interpolation. To ensure a conservative estimate of residual variance, we did not exclude non-responses [113]. We used a Continuous Decomposition Analysis (CDA) to preprocess our data. CDA decomposes SC data into tonic and phasic activity, based on a standard deconvolution method [111,114]. SCR was calculated by taking into account a baseline window set up for each participant at 500 milliseconds before the beginning of each stimulation. For a SC response to be considered as evoked by the stimulation, the phasic component had to reach a minimum threshold of 0.01 μS [114] in the 4 s after the beginning of the stimulation. For each participant, we averaged the trials for each location of stimulation then extracted three parameters: the AUC of the phasic activity (in μS.s), the maximum amplitude of response (in μS) and the latency of the maximum response (in s) calculated from the 10 s following the start of the stimulation. We analyzed the mean baseline value (in µS).

#### 2.4.4. Heart Rate

We analyzed the beat-to-beat interval (RR interval). ECG processing and analyses were performed in a MATLAB environment. Identification of QRS complex peaks from the ECG was carried out automatically by an in-house-developed detector algorithm [115]. Plots of subjects’ ECG and resulting inter-beat intervals (IBI) sequences were visually inspected, and R peaks identified by the algorithm appeared on the ECG plot to allow a quality check and interactive corrections [115]. When necessary, interpolation was done using linear interpolation [116]. The RR interval sequence were obtained by homogeneous resampling of the IBI sequence at 10 Hz [117]. The average heart rate value was removed from the RR interval sequence by subtracting the moving median computed over the previous 15 s to preserve causality [115,118]. Single trial RR interval responses were considered in a time window spanning 4 s before and 12 s after the tactile stimulus onset, then averaged by participant. From these responses, we thus extracted a RR interval variation (in s). The RR interval baseline value (in s) was extracted during the 4-s window before the beginning of the stimulation, without the moving median correction.

### 2.5. Statistical Analyses

All the statistical analyses were performed in STATISTICA^®^ (version 2020.1.2) and JAMOVI^®^ (version 2.3.9). The distribution of the data was verified using the Kolmogorov–Smirnov test and the homogeneity of the variance was calculated using the Levene test.

We used repeated measures ANOVAs. The effects of group (between: TD vs. ASD) and location (within: arm vs. hand) were tested on pupil (AUC, maximum amplitude of response, latency of maximum response, baseline value), skin conductance (AUC, maximum amplitude of response, latency of maximum response, baseline value) and cardiac parameters (RR interval variation, baseline value). To fit the parsimony principle, our strategy was the following: First, we evaluated the difference in pressure applied between the two locations (and groups when applicable), and if statistically different we used the pressure applied as a continuous predictor in our model; second, we tested the effect of age as a continuous covariate, and if statistically significant we kept age as a covariate in our model. Finally, we evaluated the effect of sex (as a categorical factor) only in the TD group analysis because of the imbalance in the ASD group, but as it was not statistically significant, we did not keep sex as a factor in our model. The repeated measures ANOVAs were evaluated within the general mixed model, and all corrections were performed within the model. In case of significant (or tendency of) interaction, post hoc tests with Bonferroni corrections for planned comparisons were performed.

We also compared the time courses of the physiological responses as a function of the site of stimulation in TD and in ASD participants. In order to do that, we downsampled the signals to 10 Hz. For each 500-ms sample, we calculated the median value for each parameter. Then, we conducted repeated measures ANOVAs to compare the two conditions of stimulation across time windows and across groups. For the pupil, 9 time points were tested (0 to 4.5 s), and for SCR and RR interval, 20 time points were tested (0 to 10 s). The *p*-value was corrected according to the Greenhouse–Geisser correction. Post hoc tests with Bonferroni corrections for planned comparisons were performed.

To compare descriptive (age) and clinical data (SSP2, IQ) between ASD and TD groups (see Appendix A, we used Mann–Whitney tests. To compare the TD group and each of the TD subgroups for a given autonomic measure, we used repeated measures ANOVAs (groups × location).

We used the non-parametric Spearman test to evaluate the correlation between the autonomic parameters and the seeking, sensibility, registration and avoiding sub-item scores from the Sensory Profile 2 questionnaire in the TD and ASD groups, and the CARS and ADOS severity score for the ASD group. 

We also used non-parametric Spearman tests to evaluate the correlations between autonomic parameters in the TD group. These analyses were completed using a principal component analysis (PCA) based on Spearman correlations with varimax rotation.

To explore the effect of groups on previously found correlations between autonomic parameters, we performed ANCOVAs with group as a categorical factor.

All results are expressed as means with standard error (SE), with effect size expressed in η^2^p (η^2^ whenever possible) for ANOVAs, and r for Spearman’s correlation and Mann– Whitney tests. All *p*-values are corrected as described previously.

## 3. Results

All the characteristics of the participants are described in Table 1 and Appendix A. As reported in the Methods section, we only obtained sufficient and analyzable physiological data in 20 out of 37 ASD children, and even less depending on the ANS effector considered. As a consequence, and because ANS reactivity to touch in TD children has not been reported before, we first described the global ANS reactivity of our total TD group (n = 30), and then, for each ANS effector, we compared the corresponding ASD group to a specific TD subgroup matched for age and number of participants.

We verified that the autonomic parameter selected for each TD subgroup (TD_Pup_ for pupil, TD_SCR_ for SCR, and TD_RR_ for RR interval) did not differ from the corresponding autonomic parameter of the total TD group, and only the TD_Pup_ group differed for some parameters (see Appendix A). 

Globally, the TD subgroups differed from the ASD groups for the verbal IQ, non-verbal IQ and SSP2 areas but not for age (see Appendix A.

### 3.1. Affective Touch Evoked an Autonomic Response in TD Children

#### 3.1.1. Pupil Discriminated Touch Locations

From 600 trials, 13 trials were discarded due to missing data or acquisition problems. Figure 2a shows the time courses of pupil dilation following tactile stimulation. As can be observed, the tactile stimulation produced a significant pupil dilation from 1 s to 3 s for both locations of stimulations (main effect of time: F_8,232_ = 18.07, *p* < 0.001, η^2^p = 0.38; post hoc comparisons between time point 1 and other time points: *p* < 0.05 for time points 3 to 7, i.e., 1 to 3.5 s), and the pupil dilation differed for the two locations of stimulation (main effect of location: F_1,29_ = 7.31, *p* = 0.01, η^2^p = 0.20), especially between 3 and 4 s (significant time × location interaction: F_8,232_ = 10.13, *p* < 0.01, η^2^p = 0.25; post hoc comparisons between hand and arm at each time point: *p* < 0.01 for time points 6 to 8, i.e., 3 to 4 s). Parameters extracted from these data are presented in Figure 2b. We found a mean AUC of 26.4 ± 19.0 mm.s for the arm and a mean AUC of 33.0 ± 25.0 mm.s for the hand, with no significant effect of location (F_1,29_ = 1.68, *p* = 0.21). The maximum amplitude of response was 0.23 ± 0.13 mm for the arm stimulation and 0.27 ± 0.15 mm for the hand stimulation, with no significant effect of location (F_1,29_ = 2.51, *p* = 0.12). However, the latency of the maximum response was significantly influenced by the location of stimulation (F_1,29_ = 12.3, *p* < 0.001, η^2^p = 0.30), with the maximum reached earlier following the arm stimulation (1.54 ± 0.8 s) than the hand stimulation (2.28 ± 0.95 s). 

We found no significant difference of pressure applied for the arm or hand stimulations (*p* = 0.13). We found no main effect of sex (AUC: F_1,27_ = 0.01, *p* = 0.91; maximum amplitude: F_1,27_ = 0.03, *p* = 0.86; latency of maximum: F_1,27_ = 0.48, *p* = 0.49), or age (AUC: F_1,27_ = 0.02, *p* = 0.96; maximum amplitude: F_1,27_ = 0.06, *p* = 0.80; latency of maximum: F_1,27_ = 0.01, *p* = 0.89). The baseline value was not affected by sex (F_1,27_ = 0.41, *p* = 0.52) or age (F_1,27_ = 0.71, *p* = 0.40).

#### 3.1.2. Skin Conductance Reacted to Pleasant and Affective Touch

No trials were discarded for the SCR analysis. As shown in Figure 2c, the tactile stimulations evoked a significant SCR from 3 s to 7 s for both locations of stimulation (main effect of time: F_19,551_ = 12.52, *p* < 0.001, η^2^p = 0.30; post hoc comparisons between time point 1 and other time points: *p* < 0.05 for time points 6 to 14, i.e., 3 to 7 s). We found no main effect of location (F_1,29_ = 0.37, *p* = 0.54), nor interaction between time and location (F_19,551_ = 1.32, *p* = 0.16). Parameters extracted from these data are presented in Figure 2d. We found a mean AUC of 15.30 ± 19.0 μS.s for the arm and of 15.4 ± 17.0 μS.s for the hand stimulation, with no significant effect of location (F_1,29_ = 5.10–5, *p* = 0.99). The maximum amplitude of response was 0.35 ± 0.47 μS for the arm and 0.36 ± 0.41 μS for the hand stimulation, with no significant effect of location (F_1,29_ = 0.01, *p* = 0.90). The latency of the maximum response was 4.18 ± 2.19 s after the beginning of the arm stimulation, and 4.53 ± 2.03 s after the beginning of the hand stimulation, with no significant effect of location (F_1,29_ = 0.37, *p* = 0.54).

We found no main effect of sex (AUC: F_1,27_ = 2.23, *p* = 0.14; maximum amplitude: F_1,27_ = 1.98, *p* = 0.17; latency of maximum: F_1,27_ = 0.02, *p* = 0.87), or age (AUC: F_1,27_ = 0.01, *p* = 0.91; maximum amplitude: F_1,27_ = 0.09, *p* = 0.76; latency of maximum: F_1,27_ = 1.04, *p* = 0.31). The baseline value was not affected by sex (F_1,27_ = 0.06, *p* = 0.79) or age (F_1,27_ = 0.08, *p* = 0.70).

#### 3.1.3. Heart Rate Decelerated during Affective Touch

In total, 12 out of 600 trials were discarded due to missing data or acquisition problems. Figure 2e shows that the RR interval varied during touch, with a significant increase between 1 and 4 s (main effect of time: F_19,551_ = 41.12, *p* < 0.001, η^2^p = 0.34; post hoc comparisons between time point 1 and other time points: *p* < 0.05 for time points 2 to 8, i.e., 1 to 4 s). We found no main effect of location (F_1,29_ = 0.31, *p* = 0.57), nor interaction between time and location (F_19,551_ = 1.65, *p* = 0.13). The tactile stimulation induced an increase of the RR interval during arm (0.03 ± 0.02 s) and hand stimulation (0.04 ± 0.02 s) compared to the baseline, i.e., we observed a deceleration of the heart rate for TD, but with no significant effect of location (F_1,29_ = 2.21, *p* = 0.14).

No effect of sex (F_1,27_ = 0.44, *p* = 0.50), nor age (F_1,27_ = 0.004, *p* = 0.95) was observed. We did find a tendency towards an age effect on the RR interval baseline values (F_1,27_ = 3.58, *p* = 0.07, η^2^p = 0.12) but no sex effect (F_1,27_ = 0.49, *p* = 0.48).

#### 3.1.4. Some Correlations within and between Autonomic Indices Parameters

We first checked correlations within each ANS effector. We found that for both pupil and SCR, the AUC significantly correlated with the maximum amplitude of response (pupil: r = 0.88, *p* < 0.0001; SCR: r = 0.96, *p* < 0.0001). Interestingly, we also found some correlation with the baseline values. For pupil size, we observed a significant negative correlation between the baseline value and the maximum amplitude of response (r = −0.43, *p* = 0.017). For the RR interval, the baseline value was significantly positively correlated with the RR interval variation (r = 0.38, *p* = 0.036).

Then, we performed correlation analyses between pupil, SCR and RR interval parameters. We only found a significant positive correlation between the RR interval baseline and the latency of the maximum response of SCR (r = 0.51, *p* = 0.004). There was no correlation between the baseline values of the three effectors (*p* > 0.43). To explore the relationships between all the parameters recorded, we performed a PCA analysis on the ten parameters (pupil: AUC, amplitude, latency, baseline; SCR: AUC, amplitude, latency, baseline; RR interval: variation and baseline) averaged for the two locations of stimulation. The first four factors accounted for 73.6% of variability. Interestingly, the first factor grouped pupil AUC, pupil maximum amplitude of response and pupil baseline; the second factor grouped SCR AUC and SCR maximum amplitude of response; the third factor grouped RR interval variability, RR interval baseline and SCR baseline; and the fourth factor grouped the latency of the maximum response for both pupil and SCR.

### 3.2. Affective Touch Evoked Smaller Autonomic Responses in ASD

Pressure applied differed between the ASD and TD subgroups (ASD_Pup_ vs. TD_Pup_: F_1,40_ = 34.27, *p* < 0.001, η^2^p = 0.46; ASD_SCR_ vs. TD_SCR_: F_1,63_ = 58.70, *p* < 0.001, η^2^p = 0.48; ASD_RR_ vs. TD_RR_: F_1,68_ = 71.34, *p* < 0.001, η^2^p = 0.51), but with no effect of location (ASD_Pup_ vs. TD_Pup_: F_1,40_ = 0.22, *p* = 0.64; ASD_SCR_ vs. TD_SCR_: F_1,63_ = 0.69, *p* = 0.40; ASD_RR_ vs. TD_RR_: F_1,68_ = 0.81, *p* = 0.36) and no interaction (ASD_Pup_ vs. TD_Pup_: F_1,40_ = 0.08, *p* = 0.77; ASD_SCR_ vs. TD_SCR_: F_1,63_ = 1.06, *p* = 0.30; ASD_RR_ vs. TD_RR_: F_1,68_ = 1.40, *p* = 0.23). Thus, we kept the pressure as a covariate in subsequent analyses.

#### 3.2.1. Pupil Dilation Tended to Discriminate TD and ASD Children

Only in 11 ASD participants could we record enough trials to consider them in an analysis, with a total of 69 trials for hand stimulation and 72 trials for arm stimulation. We selected TD children to obtain an age-matched subgroup with the same number of participants (n = 11, TD_Pup_; see Appendix A). In these TD participants, 13 out of 220 trials were discarded due to missing data or acquisition problems.

The time courses of pupil variation are presented in Figure 3a. We found no main effect of time (F_8,144_ = 1.07, *p* = 0.34), no main effect of location (F_1,18_ = 0.37, *p* = 0.54), and no main effect of pressure (F_1,18_ = 1.30, *p* = 0.27), but a tendency towards a group effect (F_1,18_ = 3.30, *p* = 0.08, η^2^p = 0.15). We also observed a tendency towards the interaction between time × location × group (F_8,144_ = 2.37, *p* = 0.08, η^2^p = 0.11), but no other significant interaction. As we already described the effect of location over time within the TD group (see Section 3.1.1), we focused on the comparisons between TD and ASD children for each location of stimulation, and on the comparisons between the two locations of stimulation within the ASD group. Post hoc comparisons revealed a difference between TD and ASD participants following hand stimulation (*p* = 0.05 for time point 7, i.e., 3.5 s).

Parameters extracted from pupil data are presented in Figure 3b. The AUC did not differ between TD_Pup_ and ASD_Pup_ (F_1,18_ = 0.84, *p* = 0.36) and we did not find an effect of location (F_1,18_ = 0.001, *p* = 0.96). We found a tendency towards the interaction between location × group (TD_Pup_ hand: 42.4 ± 29.9 mm.s, TD_Pup_ arm: 20.2 ± 14.3 mm.s; ASD_Pup_ hand: 20.2 ± 14.3 mm.s, ASD_Pup_ arm: 24.9 ± 38.0 mm.s; F_1,18_ = 3.29, *p* = 0.08, η^2^p = 0.20), but with no significant post hoc comparison (*p* > 0.16). No effect of pressure was found (F_1,18_ = 0.52, *p* = 0.47). The maximum amplitude of response did not differ between TD_Pup_ and ASD_Pup_ groups (F_1,18_ = 0.39, *p* = 0.54), nor between locations (TD_Pup_ hand: 0.32 ± 0.17 mm, TD_Pup_ arm: 0.21 ± 0.14 mm; ASD_Pup_ hand: 0.24 ± 0.19 mm, ASD_Pup_ arm: 0.19 ± 0.26 mm; F_1,18_ = 0.12, *p* = 0.72), or pressure (F_1,18_ = 0.49, *p* = 0.49). No significant interaction was found. However, the latency of the maximum response tended to be influenced by the group (F_1,18_ = 3.69, *p* = 0.07, η^2^p = 0.17), and by the location (TD_Pup_ hand: 2.77 ± 0.77 s, TD_Pup_ arm: 1.55 ± 0.9 s; ASD_Pup_ hand: 1.52 ± 1.50 s, ASD_Pup_ arm: 1.59 ± 1.40 s; F_1,18_ = 3.95, *p* = 0.06, η^2^p = 0.18). The interaction between these two factors was significant (F_1,18_ = 4.49, *p* = 0.04, η^2^p = 0.20) with statistical differences between TD_Pup_ and ASD_Pup_ hand conditions (post hoc: *p* = 0.056) and TD_Pup_ hand and arm conditions (post hoc: *p* = 0.03). No effect of pressure was found (F_1,18_ = 0.05, *p* = 0.08) or other significant interaction. 

We found no main effect of age (AUC: F_1,17_ = 0.79, *p* = 0.38; maximum amplitude: F_1,17_ = 0.32, *p* = 0.57; latency of maximum response: F_1,17_ = 0.46, *p* = 0.50).

#### 3.2.2. Skin Conductance Response Was Not Different between ASD and TD Children

Due to excessive hand movements, we discarded one ASD participant. The resulting ASD_SCR_ group was composed of 19 children, and was compared to the TD_SCR_ age-matched group (which did not differ from the total TD group; see Appendix A.

The time courses of SCR are presented in Figure 3c. We found no main effect of time (F_19,532_ = 1.70, *p* = 0.18), no main effect of location (F_1,28_ = 0.35, *p* = 0.55), no main effect of pressure (F_1,28_ = 0.02, *p* = 0.86), and no main effect of group (F_1,28_ = 1.15, *p* = 0.69). We observed a tendency towards the interaction between location × group (F_1,28_ = 3.66, *p* = 0.06, η^2^p = 0.11), but with no significant post hoc comparison (*p* > 0.20). No other significant interaction was observed (*p* > 0.40) 

Parameters extracted from SCR data are presented in Figure 3d. The AUC did not differ between TD_SCR_ and ASD_SCR_ (F_1,29_ = 0.13, *p* = 0.71) and we did not find an effect of location (F_1,29_ = 0.07, *p* = 0.78), but we found a tendency towards the interaction between location × group (TD_SCR_ hand: 15.2 ± 17.9 µS.s, TD_SCR_ arm: 9.88 ± 9.47 µS.s; ASD_SCR_ hand: 11.1 ± 9.13 µS.s, ASD_SCR_ arm: 15.3 ± 13.8 µS.s; F_1,29_ = 3.36, *p* = 0.07, η^2^p = 0.10), but with no significant post hoc comparison (*p* > 0.16). No effect of pressure was found (F_1,29_ = 0.74, *p* = 0.39). The maximum amplitude of response did not differ between TD_SCR_ and ASD_SCR_ groups (F_1,29_ = 0.14, *p* = 0.70), nor between locations (F_1,18_ = 0.01, *p* = 0.92). We found a tendency towards the interaction between location × group (TD_SCR_ hand: 0.33 ± 0.39 µS, TD_SCR_ arm: 0.20 ± 0.22 µS; ASD_SCR_ hand: 0.25 ± 0.19 µS, ASD_SCR_ arm: 0.29 ± 0.28 µS; F_1,29_ = 3.48, *p* = 0.07, η^2^p = 0.10), but with no significant post hoc comparison (*p* > 0.20). No effect of pressure was found (F_1,29_ = 0.92, *p* = 0.34). The latency of the maximum response showed no effect of the group (F_1,29_ = 0.53, *p* = 0.47), no effect of location (F_1,29_ = 2.38, *p* = 0.13) and no interaction (TD_SCR_ hand: 4.78 ± 1.81 s, TD_SCR_ arm: 3.82 ± 2.08 s; ASD_SCR_ hand: 4.65 ± 1.96 s, ASD_SCR_ arm: 4.44 ± 2.68 s; F_1,29_ = 0.06, *p* = 0.79). No effect of pressure was found (F_1,29_ = 0.02, *p* = 0.87), nor other significant interaction (*p* > 0.45).

We found a main effect of age on AUC (F_1,28_ = 5.59, *p* = 0.02, η^2^p = 0.16), on maximum amplitude (F_1,28_ = 4.01, *p* = 0.05, η^2^p = 0.12), and a tendency towards an age effect on the latency of the maximum response (F_1,28_ = 3.60, *p* = 0.06, η^2^p = 0.11).

#### 3.2.3. RR Interval Did Not Increase following Pleasant Stimulation in ASD Children

For RR interval analyses, the ASD_RR_ group was composed of 20 participants. We selected the 20 TD children to compose the age-matched TD_RR_ group, which did not differ from the total TD group (see Appendix A. We discarded 12 trials for TD participants due to body movements which impaired the quality of the signal.

The time courses of the RR interval variation are presented in Figure 3e. We found a main effect of time (F_19,551_ = 5.16, *p* < 0.01, η^2^p = 0.16), a main effect of location (F_1,29_ = 4.55, *p* = 0.04, η^2^p = 0.13), no main effect of pressure (F_1,29_ = 1.55, *p* = 0.22), but an effect of the group (F_1,29_ = 7.11, *p* = 0.01, η^2^p = 0.19). We observed an interaction between time × group (F_19,551_ = 4.63, *p* < 0.01, η^2^p = 0.13), but no other significant interaction (*p* > 0.74). Post hoc comparisons revealed a difference in the TD_RR_ group between time point 1 and time points 3 to 7 (i.e., 1.5 to 3.5 s) and between time point 1 and time points 18 to 20 (i.e., 9 to 10 s) (*p* < 0.046). No significant comparison was present within the ASD_RR_ group (between time point 1 and other time points; *p* > 0.74), nor between TD_RR_ and ASD_RR_ groups (comparison at each time point; *p* > 0.17).

The parameter extracted from the RR interval data is presented in Figure 3f. The RR interval variation during tactile stimulation differed between TD_RR_ and ASD_RR_ (F_1,29_ = 6.02, *p* = 0.02, η^2^p = 0.17) but we did not find an effect of location (F_1,29_ = 0.01, *p* = 0.91) nor an interaction between group and location (TD_RR_ hand: 0.04 s ± 0.02 s, TD_RR_ arm: 0.03 ± 0.02 s; ASD_RR_ hand: 0.02 ± 0.02 s, ASD_RR_ arm: 0.01 ± 0.02 s; F_1,29_ = 0.65, *p* = 0.42). No effect of pressure was found (F_1,29_ = 0.20, *p* = 0.65) and we did not find any other significant interaction (*p* > 0.37). We found no effect of age on the RR interval variation during tactile stimulation (F_1,28_ = 1.87, *p* = 0.18).

### 3.3. Baseline Values Were Different between TD and ASD Children

Baseline values are presented in Figure 4. The baseline value of pupil size was bigger for TD_Pup_ (6.09 ± 0.66 mm) compared to ASD_Pup_ (5.18 ± 0.85 mm; F_1,19_ = 7.55, *p* = 0.013; η^2^ = 0.28) and no effect of age was found (F_1,18_ = 1.26, *p* = 0.27). The baseline value of SCR was smaller for TD_SCR_ (4.10 ± 3.07 µS) compared to ASD_SCR_ (5.64 ± 3.71 µS) but there was no statistical difference (F_1,36_ = 2.34, *p* = 0.13), nor an effect of age (F_1,35_ = 0.57, *p* = 0.45). We also found a significant effect of group on the RR interval baseline values (TD_RR_: 0.751 ± 0.11 s; ASD_RR_: 0.633 ± 0.09 s; F_1,38_ = 13.3, *p* < 0.001, η^2^ = 0.26) but no effect of age (F_1,37_ = 0.20, *p* = 0.65).

### 3.4. Baseline Differences Did Not Explain Affective Touch Reactivity Differences between TD and ASD Children

Considering the number of ASD children for which we obtained data for the three ANS effectors, we did not explore correlations between the autonomic parameters. However, as we found (1) a correlation between the pupil baseline value and the pupil maximum amplitude of response, and the RR interval baseline value and the RR interval variation in TD children, and (2) a significant difference in the baseline values for pupil and RR interval between TD and ASD children, we explored the effect of group on the relationship between baseline values and the response to stimulation for both ANS effectors. We performed an ANCOVA with a categorical factor (ASD vs. TD). For pupil, we found no significant regression between the baseline value and the maximum amplitude of response when controlling for the group (F_1,18_ = 0.21, *p* = 0.65). For RR interval, we found no significant regression between the baseline value and the RR interval variation when controlling for the group (F_1,37_ < 0.0001, *p* = 0.99).

### 3.5. Few Correlations between Autonomic and Clinical or Sensory Parameters

We explored the correlations between our autonomic parameters and the scores of the four areas of the parent- or therapist-filled SSP2. In the total TD group, TD subgroups and ASD groups, no significant correlation was found between pupil, SCR, or RR interval parameters and the four areas scores of the SSP2 (*p* > 0.05).

For children with ASD, we also explored the correlations between our autonomic parameters and the ADOS-2 severity and CARS scores. We found no significant correlation between the ADOS-2 severity score and the pupil parameters (*p* > 0.05), except for the latency of the maximum response (r = 0.65, *p* = 0.04). We found no significant correlation between the ADOS-2 severity score and SCR parameters (*p* > 0.05) or RR parameters (*p* > 0.05). We found no significant correlation between the CARS and the pupil (*p* > 0.05) or SCR parameters (*p* > 0.05), but the RR interval variation negatively correlated with the CARS scores (r = −0.54, *p* = 0.02; not the RR interval baseline: *p* = 0.59).

To exclude any effect from IQ scores in observed differences between TD and ASD children, we checked that our autonomic parameters were not correlated to these scores. In the total TD group, for which we had both verbal and perceptive scores, no correlations were observed (*p* > 0.05). In ASD children, we could only obtain verbal scores in enough children to test for correlations, and no correlations were observed (*p* > 0.05).

## 4. Discussion

### 4.1. Autonomic Responses in TD Children

In TD children, the overall pattern of results showed that an affective tactile stimulation elicited both a response from the SNS, with a dilation of the pupil and an elevation of the SCR, and PNS, with a deceleration of the heart rate (an increase of the RR interval). This pattern corresponds to what could be expected from the literature, but our study is the first to show these three simultaneous reactions in children and for tactile stimuli. 

Our SNS responses are coherent with what has been described in adults by van Hooijdonk et al. [57], who also reported a pupil dilation and a SCR elevation following a pleasant affective-touch stimulation on the palm of the hand or the dorsal face of the hand. However, they showed that their responses were not related to CT fiber specific stimulation, but rather to the intensity of their stimulation, with larger responses for higher velocities (and thus less pleasant). The site of stimulation did not affect their responses. On the contrary, Bertheaux et al. [56], in a texture exploration context, showed that pupil dilation was modulated by the perceived emotional intensity. In our study, we used the same velocity for our two sites of stimulation, and thus could not contrast pleasant and less pleasant stimulations. Our design was not aimed at evaluating the effect of stimulation intensity, but we did not find any influence of pressure applied on our responses. While we observed a small effect on pupil time courses, the maximum amplitude and AUC for both pupil dilation and SCR were not affected by the location of stimulation (CT high- vs. low-density territories), contrary to our hypothesis. Considering that Olausson et al. [119] have shown, in two neuronopathy subjects with only CT fibers, that stroking the forearm induced an elevation of the SCR, we expected a larger pupil dilation and SCR for our CT high-density stimulation. Our results thus do not support a specific role of CT fibers in the valuation of affective touch, as least visible in autonomic responses.

However, we found a difference between our two sites of stimulation on the latency of the pupil maximum response. Whereas the dilation of the pupil started with the same dynamics for the two stimulations, the stimulation of the hand (CT low-density territory) led to a later maximum of dilation. One possible factor that could contribute to a difference in response length could be the adaptation of the mechanoreceptors: both in the hand [120] and in hairy skin [121], receptors with myelinated fibers are a mix of slow and fast adaptation receptors, while CT fibers have an intermediate adaptation [122]. It is thus possible that stimulating the palm of the hand, with a large quantity of slow adaptation receptors, could elicit a longer neuronal discharge. Another possible interpretation could be the behavioral repertoire evoked by the stimulation, or direction of motivation [54], with a hand stimulation more likely to prompt us to act [123].

The cardiac deceleration we observed is coherent with many reports, such as Fairhurst et al. [60] in infants or Valenza et al. [124] in adults. CT-optimal stimulation leads to heart rate deceleration, with an increased behavioral engagement, while slow or fast non-optimal stimulations lead to heart rate acceleration [60]. Pleasant affective stimulation would thus evoke empathy and engagement responses, as proposed by Porges’ Polyvagal theory [70,71]. The HR changes we observed lasted the duration of the stimulation, followed by a return to the baseline. It was thus not possible to test the hypothesis proposed by Walker et al. [20], of a triphasic cardiac waveform that would reflect an initial orienting response followed by recognition processes. We did not observe any difference as a function of the stimulation site, as described by Pawling et al. [65].

This coactivation of SNS and PNS parameters has already been described in the context of viewing emotional pictures or faces [55,79], illustrating the possibility of more than antagonist functioning between the SNS and PNS [68,69]. However, the responses of the different ANS effectors we measured were not correlated with each other, even for the two SNS responses that are pupil dilation and SCR. We have described pupil dilation as a sympathetic parameter, as indeed the SNS controls the mobilization of the iris dilator muscle. However, contrary to the SCR that is under strict SNS control, pupil diameter is also influenced by the PNS. A longer latency of pupil dilation may thus also reflect a PNS inhibition, that would not influence the SCR. The pupil seems to be a very sensitive measure, the only one showing a difference as a function of our stimulation location, integrating many emotional and cognitive inputs [77].

### 4.2. Differences between TD and ASD Children

Globally, we observed smaller or absent autonomic responses to affective touch in ASD children compared to TD children, without any effect of the location of stimulation, as well as different baseline values for pupil diameter and heart rate.

While we could test the autonomic response to affective touch only in a small sample of ASD children, we found some differences with TD children. For the pupil diameter, the time course analysis showed no significant dilation during the stimulation, contrary to what was observed in TD children. The specific effect of stimulation site on the latency of the maximum response was not present either. This absence of effect was probably due to the overall lower reactivity in ASD children rather than to an effect of specific skin location or recruited tactile fibers. The lower pupil reactivity to social and emotional stimuli has only been documented for face observation in ASD children [93]. The SCR results for ASD children are less clear, with no statistical differences between the TD and ASD groups, but no difference with the baseline either. We have to acknowledge a variability in the SCR responses, even within the TD groups. Indeed, while the total TD group (n = 30) showed no difference for the two locations of stimulation, the TD_SCR_ group (n = 19) seemed to exhibit a different profile (Figure 3c,d), even if not statistically significant. To conclude on the SCR results, a larger sample of children may be necessary (see also Section 4.3), possibly to account also for the individual variability of SCR responses, with individuals being non-responders and others being active responders. In a rest protocol, we recently showed that 6–12-year-old children showed even more variability in the number of spontaneous SCR events than adults [125]. Finally, we found no difference in the RR interval variation from baseline during tactile stimulation in ASD children, contrary to the TD group. This result could be interpreted within the Porges’ theory [70,71] context as the tactile stimulation evokes neither an affiliative and social engagement behavior, nor a stressful reaction. It may be viewed as a reduced engagement response, possibly linked to motivation withdrawal observed in ASD [126] or non-rewarding interpersonal touch experiences [127].

One study had previously suggested that, peripherally, individuals with ASD may have a smaller density of CT fibers than TD individuals [128]. If that is the case, our results suggest that autonomic responses to affective touch, when present, may reflect the overall percept of pleasantness (preserved in ASD adults, even if with more extreme judgements of pleasantness and unpleasantness; [33,39]) and not the nature of fibers stimulated. Cascio et al. [40] also suggested that ASD children found affective touch stimulation less pleasant than their TD peers did. Such a difference in percept might account also for our lower autonomic responses in ASD children.

Our results also revealed that ASD children had altered baseline values for pupil diameter and heart rate compared to age-matched TD children. Baseline pupil diameter in ASD is a topic with conflicting results, with some studies reporting a smaller pupil size in ASD than TD children (e.g., [92]) like in the present study, others reporting a larger pupil size in ASD (e.g., [129]), and others finding no difference (for review, see [130]). Pupil diameter has been shown to reflect arousal and locus coeruleus discharge [131], thus pointing towards a lower arousal in ASD. However, increased heart rate was repeatedly found in children with ASD compared to TD controls under resting conditions, as well as during mental stress (e.g., [85,99]). This altered cardiovascular pattern seems to be predominantly mediated by the PNS. Specifically, the reduced cardiovagal control at rest and in response to stress, along with the lower cardiac sensitivity, could be related to an impaired baroreceptor reflex in ASD populations [132]. It could also be speculated that basal heart rate is influenced by the resonance between respiration and the intrinsic heart system [133,134].We did not find any difference in baseline skin conductance, as reported by Keith and al. [135], and contrary to previous studies reporting higher [85] or lower [88] SCR baseline values. Other sympathetic parameters, like the pre-ejection period, have also been found as not affected in ASD children compared to TD children [136,137]. Overall, these results show that the chronic hyper-arousal described in autism for decades may be linked to the autonomic parameter measured (for review, see [87]): when considering parameters reflecting PNS activity, hyper-arousal (i.e., lower PNS control, as exemplified by the higher heart rate) is observed, while other autonomic parameters may suggest hypo-arousal (e.g., lower basal pupil diameter). This could reflect a disharmony in the autonomic mobilization of the different physiological organs. However, we want to be careful in comparing our baseline values to what has been described in rest protocols: what we described in this study are the values taken by our autonomic parameters in between the stimulations, and these values may also reflect engagement (or non-engagement) in the protocol, global arousal due to tactile stimulations, etc.

An important result is that, in ASD children, the baseline values did not predict the responsivity to tactile stimulation. Contrary to what we described in TD children, for whom a lower baseline pupil diameter or heart rate were associated with larger responses to the tactile stimulation, this relationship disappeared when considering the two groups of children (ASD and TD). Our reduced autonomic reactivity to affective touch in ASD was thus not explained by an underlying dysautonomia.

One of our hypotheses was that autonomic reactivity to the sensory stimulation would reflect the sensory anomalies described by questionnaires. However, we found no correlation between the SSP2 scores and any of our autonomic parameters, in TD and ASD children. One possible reason is that neurophysiological features contribute only marginally to the sensory symptoms captured by clinical assessments and may instead reflect differences in information processing more directly. These results are consistent with what has been described by Güçlü et al. [31], supporting the hypothesis that there is probably no causal relationship between qualitative features identified by questionnaires and neurophysiological features. In addition, note that, in children, SSP2 scores are calculated based on parents’ reports, which could even lower a possible correlation between these measures. Our results, however, do not mean that neurophysiological measures do not contribute to explain a common core feature of ASD, as is the altered tactile responsivity. Rather, it may be that clinical sensory profiles are best suited to find associations with aspects of touch anchored in high-level social contexts. This absence of correlation between physiological and behavioral parameters confirm what has already been described in a few studies [95,96] and underline the potential clinical importance of measuring physiological parameters to complement the sensory profile of ASD individuals. Interestingly, we did not find correlation between the CARS or ADOS-2 severity scores and SSP2 scores in ASD individuals, or between the CARS or ADOS-2 severity scores and the autonomic parameters (except between ADOS-2 severity and pupil latency of maximum response, and between CARS and the RR variation). As we aimed at recruiting ASD children on the whole spectrum, our ASD group also differed from the TD children in their IQ levels, but IQ scores were not correlated to autonomic parameters either. These absences of correlations suggest that, even though our sample of tested ASD children could not include all ASD participants with severe autism or severe developmental delay, our physiological results may be representative of the overall ASD population. Previous imaging studies had, however, described correlations between autism scores and cerebral activations during affective touch (e.g., [41]). The discrepancy with our results may be due to the fact that autonomic responses integrate many cortical and subcortical inputs, and may not be sensitive enough to pick up on the specific patterns exhibited in a few cortical regions.

### 4.3. Limits of the Study

The main limitation of our study is the number of ASD children we managed to record. Our hope was that peripheral autonomic measures would be well tolerated by a large part of the autism spectrum, and would allow us to explore tactile perception, which is less studied than other sensory stimulations. A combination of two factors led to the fact that nearly half the ASD children for whom we obtained consent failed to complete the protocol. The first one was the novelty of the room and of the recording devices. For children who visited the hospital regularly, we could habituate them to the material and context, but others came only once, and while willing to participate, could not proceed. The second factor was the tactile experiment itself. In the context of a visual experiment (e.g., [93]) or a passive auditory experiment without pupil recordings during which children can look at a silent movie (e.g., [138]), the screen displays an engaging visual stimulation that helps to focus the children. In our experiment, as we wanted to record the pupil response to tactile stimulations, no such visual stimulation was used, and ASD children in particular were easily distracted and looked at the tactile stimulation. As a result, very few ASD children could be recorded, especially for pupil diameter. The ASD children that could not be recorded exhibited overall more severe autism, more developmental delay and more atypical sensory scores. However, as the autonomic measures did not correlate with these clinical scores, we could expect similar physiological results for this portion of the autism spectrum. Our results will nonetheless need to be replicated in a larger population, also taking into account the difference in gender ratio between ASD and TD groups. A larger population would also allow us to perform analyses to estimate possible age effects within the age range we tested, such as regression or cluster analyses. Indeed, some autonomic parameters we measured, such as heart rate, are known to be influenced by age (e.g., [125]).

In order to facilitate the interaction with children, we also chose not to use an automated system for stimuli deliverance. While very reliable, since they depend neither on the participants’ interaction with the object [56] nor the researcher presenting the stimulus [57,139], these systems are rather impressive and require the participants not to move at all. Our manual stimulation was thus more adapted for ASD children, and the experimenter could wait for calm periods to continue with the protocol if necessary. However, it could introduce human flaws, and in the end, we experienced a difference in the pressure applied between our two groups of children (even if our statistical analyses showed no effect of this factor on our results). These technical difficulties also account for tactile perception being less studied overall: tactile stimuli are harder to deliver and to monitor.

Affective touch has been studied with several approaches: variation of the stimulus velocity, to target optimally or non-optimally CT fibers (e.g., [45,57]); variation of the pleasantness of the texture used (e.g., [39]); and variation of the location of the stimulation (e.g., [41]), as we did in our study. As a consequence, and as our results mainly showed no effect of the location of the stimulation, we cannot conclude definitely on the involvement of CT fibers in the autonomic responses we described. We wanted to keep the study as short as possible, not to overburden ASD children, and also did not vary stimulation velocity [57]. A larger design, with variation of both the location and the velocity of stimulation, may be required to explore CT fibers’ role in the autonomic responses we described in TD children. It would also be beneficial to precisely estimate the pleasantness of our stimulations, at least in TD children. As Shirmer and McGlone [140] claim, affective touch goes beyond simple stroking of hairy skin, and our results may be linked to the hedonic value of our stimulations. In the present study, our aim was to test a protocol applicable in children with ASD, that could have a low intellectual quotient and no language, and this assessment was not possible. 

Finally, in order to go further in the understanding of autonomic regulation, we could have added other indices. In particular, we have interpreted RR interval as regulated by the PNS. However, heart rate is not uniquely explained by a parasympathetic vagal control, as it can also be influenced by an inhibition of the sympathetic innervation of the heart. Other measures, such as the heart rate variability (HRV), pre-ejection period (PEP) and the respiratory sinus arrhythmia (RSA), are more suited to explore the sympathetic tone and the cardiac vagal tone, respectively, but are not appropriate to explore the physiological response to a short sensory stimulation due to their timing requirements (see [124] for recent technological developments to estimate heart rate variability in response to short events). Nevertheless, our results and previous studies [97,141] suggest an activation of the nucleus ambiguous following a pleasant touch. According to Porges’ polyvagal theory [70,71], this parasympathetic activation correlates with empathy and social engagement. The opposite possibility would be a parasympathetic activation via the dorsal motor vagal branch, solicited in extreme danger conditions and leading to a freezing behavior, but it would not be coherent with studies showing a link between heart rate deceleration and both pleasantness ratings [142] and behavioral engagement [60].

## 5. Conclusions and Perspectives

In conclusion, our study shows that a pleasant affective tactile stimulation evokes a complex pattern of autonomic activations in TD children, with both sympathetic and parasympathetic responses. This autonomic reactivity is reduced in ASD children, possibly in connection with the specific percept evoked by the affective stimulation or to more general decreased arousal. Social/affective touch processing has been an area of interest in autism as it plays a fundamental role in early social interactions and in socioemotional development. Though some studies have provided evidence of altered social touch perception in autism, to our knowledge it had never been approached via autonomic reactivity in several ANS effectors recorded simultaneously. The complex autonomic pattern is best apprehended by the recording of multiple parameters, that can reveal subtle differences within the two populations and between different contexts, such as sensory reactivity and rest. The differences observed between the TD and ASD children need to be further explored to see how they could be related to temperament or behavioral strategies depending on the context.

Research on affective touch has made considerable progress in recent years (see the themed issue on Body–brain interactions/affective touch in Current Opinion in Behavioral Sciences, 2022), and it has become increasingly clear that (1) affective touch means more than C-LTMR firing; (2) CT fibers interface with sensory, cognitive and social brain circuits; and (3) these processes critically interrelate with feedback and feedforward stress-regulatory ANS circuits [20]. With regard to this field, our study is the first to describe autonomic responses in children with multiple parameters and to show autonomic disengagement in ASD in response to affective and pleasant touch.

Complementary information, such as hedonic evaluation, anxiety and stress measurements (for example via cortisol samples), would allow us to go further with the interpretation and exploration of the present results, but also to compare affective tactile stimulations with other sensory stimulations, tactile or not, social or not.

## Figures and Tables

**Figure 1 jcm-11-07146-f001:**
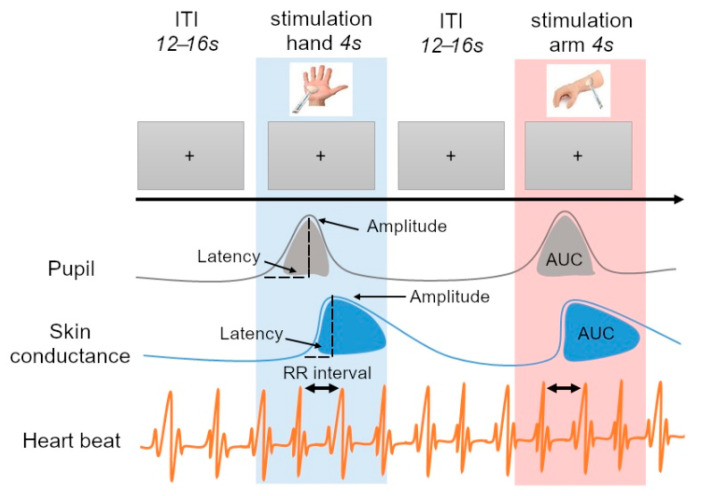
Experimental Procedure. During the experiment, pupil size, skin conductance and cardiac frequency were recorded while 10 stroking stimulations on the arm and 10 on the palm of the hand were performed by the experimenter. The stimulation phase lasted 4 s. It was preceded by a 4 s pre-stimulation phase, considered as baseline for skin conductance and cardiac frequency, and the last 250 ms of which were used to calculate the pupil baseline values. The inter-trial interval (ITI) lasted between 12 and 16 s.

**Figure 2 jcm-11-07146-f002:**
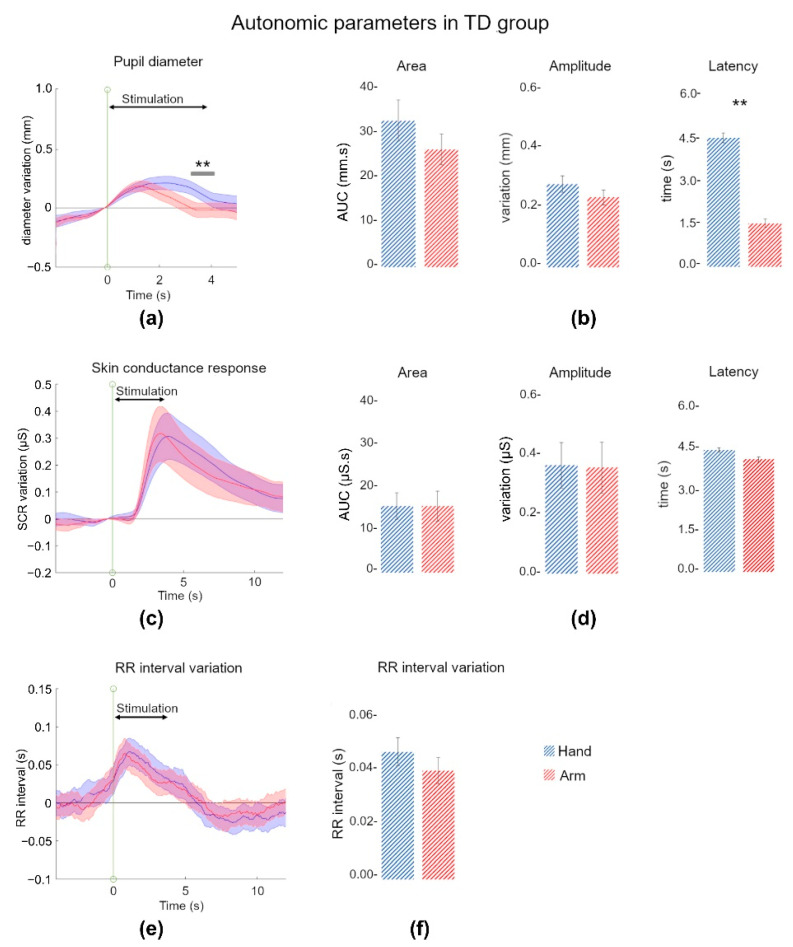
ANS responses to tactile stimulation in the total TD group. (**a**) Mean pupil dilation (relative to the baseline value, in mm) as a function of time during an arm stimulation (red) or a hand stimulation (blue). The shading around the curve corresponds to the SEM. The green vertical line (0 s) indicates the beginning of the stimulation. The continuous black line indicates the length of the stimulation. The continuous grey line indicates the significant difference between the two curves. (**b**) Mean values (±standard error) for the three pupil parameters extracted for the two stimulation conditions (arm in red, hand in blue): AUC (in mm.s), maximum amplitude (in mm), and latency of the maximum response (in s). (**c**) Mean SCR (relative to the baseline value, in µS) as a function of time during an arm stimulation (red) or a hand stimulation (blue). Same conventions as in (**a**). (**d**) Mean values (±standard error) for the three skin conductance parameters calculated for the two conditions (arm in red, hand in blue): AUC (in µS.s), maximum amplitude (in µS), and latency of the maximum response (in s). (**e**) Mean RR interval (relative to the baseline value, in s) as a function of time during an arm stimulation (red) or a hand stimulation (blue). Same conventions as in (**a**). (**f**) Mean value (±standard error) of the RR interval (in s) calculated for the two conditions (arm in red, hand in blue). **: *p* < 0.001.

**Figure 3 jcm-11-07146-f003:**
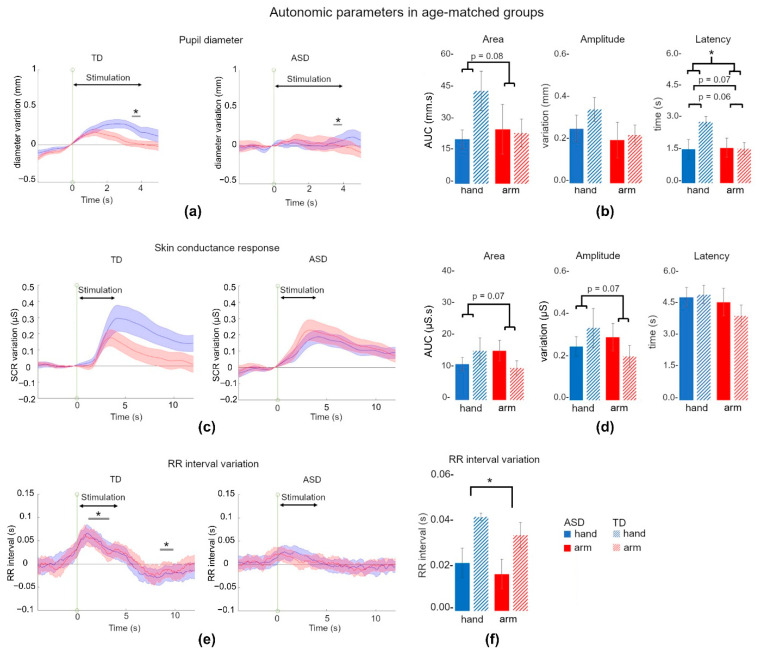
ANS responses to tactile stimulation in ASD and age-matched TD groups. (**a**) Mean pupil dilation (relative to the baseline value, in mm) as a function of time during an arm stimulation (red) or a hand stimulation (blue), in TD and ASD groups. The shading around the curve corresponds to the SEM. The green vertical line (0 s) indicates the beginning of the stimulation. The continuous black line indicates the length of the stimulation. The continuous grey line indicates the significant difference between the two curves. (**b**) Mean values (±standard error) for the three pupil parameters extracted for the two stimulation conditions (arm in red, hand in blue): AUC (in mm.s), maximum amplitude (in mm), and latency of the maximum response (in s). TD are represented with hatched columns (right columns), ASD with filled columns (left columns). (**c**) Mean SCR (relative to the baseline value, in µS) as a function of time during an arm stimulation (red) or a hand stimulation (blue), in TD and ASD groups. Same conventions as in (**a**). (**d**) Mean values (±standard error) for the three skin conductance parameters calculated for the two conditions (arm in red, hand in blue): AUC (in µS.s), maximum amplitude (in µS), and latency of the SCR (in s). Same conventions as in (**b**). (**e**) Mean RR interval (relative to the baseline value, in s) as a function of time during an arm stimulation (red) or a hand stimulation (blue), in TD and ASD groups. The shading around the curve corresponds to the SEM. Same conventions as in (**a**). (**f**) Mean value (±standard error) of the RR interval variation (in s) calculated for the two conditions (arm in red, hand in blue) and for the two groups. Same conventions as in b. *: *p* < 0.05. *p* values for tendencies are indicated.

**Figure 4 jcm-11-07146-f004:**
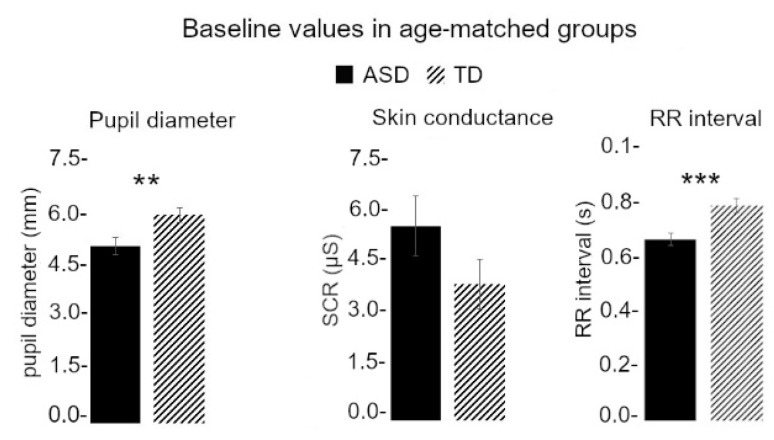
Baseline values of ANS parameters in ASD and age-matched TD groups. The figure indicates differences in the baseline pupil median diameter (mm) for the ASD (black/left columns) and TD age-matched group (hatched/right columns), baseline SCR (µS) and baseline RR interval (s). Results are expressed in mean value (±standard error). **: *p* < 0.01, ***: *p* < 0.001.

**Table 1 jcm-11-07146-t001:** Demographic and clinical data for TD and ASD groups, and ASD non-tested group. Values are expressed in mean ± standard deviation (age) and mean with range (minimum and maximum values) for clinical data. n: number of children for whom the scores could be collected. Statistical differences between the ASD and ASD non-tested groups are reported. *: *p* < 0.05, **: *p* < 0.01. See Appendix A for detailed statistical results and Appendix A for SSP2 differences. ADI-R: Autism Diagnostic Interview-Revised; ADOS: Autism Diagnostic Observation Schedule; CARS: Childhood Autism Rating Scale; IQ: Intellectual Quotient; SSP2: Short Sensory Profile 2.

	TD Group	ASD Group	ASD Non-Tested Group
n	30	20	17
sex M:F	14:16	19:1	13:4
age in years (±SD)	9.20 ± 0.20	9.13 ± 1.69	9.80 ± 1.01
verbal IQ [range]	124 [88–155]	73 [25–95] n = 13	44 [20–74] ** n = 12
non verbal IQ [range]	116 [88–146]	71 [66–129] n = 4	67 [22–121] n = 7
IQ total < 70	0	2	9 *
CARS [range]		30 [24–40] n = 17	34 [30–40] ** n = 15
ADI-R:			
- Social interaction [range]		17 [5–29] n = 17	19 [10–28] n = 13
- Verbal communication [range]		15 [6–22] n = 9	17 [10–21] n = 5
- Non verbal communication [range]		9.5 [6–14] n = 13	10.5 [7–17] n = 8
- Repetitive and restricted behaviours [range]		4.5 [1–9] n = 17	5.5 [0–12] n = 13
ADOS-2:			
- Severity score [range]		6.5 [1–10] n = 15	8 [3–10] n = 12
SSP2:			
- Avoiding [range]	29 [18–49]	53 [33–68] n = 14	55 [36–65] n = 6
- Registration [range]	26 [21–41]	40 [12–80] n = 14	42 [17–55] n = 6
- Sensibility [range]	23 [18–35]	40 [26–54] n = 14	55 [30–65] * n = 6
- Research [range]	20 [13–33]	40 [28–58] n = 14	55 [35–72] * n = 6

## Data Availability

The data presented in this study are available on request from the corresponding author. The data are not publicly available due to absence of ethical consent from the participants and their parents.

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
