# Peer review of "Atypical Response to Affective Touch in Children with Autism: Multi-Parametric Exploration of the Autonomic System"

_jcm, 2022, doi:10.3390/jcm11237146_

Round 1
Reviewer 1 Report
General Comments and Overall Evaluation (with limitations and strengths): Turning first to the general comments, this study is relevant to the mission of this journal and presents original data. The authors evaluated the autonomic response to pleasant affective touch in children with autism (n =20) and typically developing peers (n =30) matched by age. Three autonomic indices (pupil dilation, skin conductance, and heart rate) were recorded, and identical stimulations at two locations (i.e., CT high- and low-density) were compared. Overall, ASD children exhibited reduced autonomic responses and different ANS baseline values compared to TD children. Results also showed that the atypical autonomic responses to pleasant touch in children with autism were not specific to CT-fibers stimulation. However, some issues mitigate the impact of this manuscript (see below for more details), and there are several considerations that I believe need to be addressed before the study can be considered for publication.
Comments:
- Page 5, lines 218 – 221: the third objective of this study related to the sensory profiles measured by questionnaires needs a well-crafted theoretical background.
- The authors measured heart rate, but why not Heart Rate Variability?
- The sample size is quite small. Readers now expect to see a larger number of study participants, especially given the heterogeneity of symptoms in autism. Therefore, in my opinion, without a larger sample that allows robust statistical analyses, it will be difficult to make conclusions regarding this topic.
- The authors describe the developmental characteristics of the participants (e.g., nonverbal IQ…) in the supplementary material. However, it is crucial to detail the main characteristics in the Method (Participants section). Given the heterogeneity of symptoms in autism, studies with variable samples (although with the same diagnosis) lead to inconsistent findings. Thus, it is crucial to understand confound variables, and the subgroups within the spectrum need to be specified experimentally. For instance, is crucial to understand how many children have intellectual disabilities, the level of autism symptom severity level… Furthermore, in this study, only children with autism who accepted to sit on the armchair, tolerated the physiological captors, and stayed calm during the data acquisition were considered for the analyses, probably biasing the sample towards a particular sensory profile. Thus, this sample is not representative of this population. We can have the same diagnosis, but quantitative differences can be obvious.
- Diagnosing ASD can be difficult, since there is no medical test, like a blood test, to diagnose the disorder. Therefore, we should use and report the comprehensive assessment for diagnosis of ASD, including the use of validated diagnostic instruments. Indeed, the clinical diagnosis is most valid and reliable when applying standardized diagnostic tools such as ADOS-2 and ADI-R. Thus, I suggest that the authors present more details about the diagnosis (mentioning the DSM-5 and CIM-10 criteria is not enough).
- The Children Sensory Profile 2 should be described in the Material (including their psychometric characteristics)
- Page 5, line 228: How the authors verified that the TD children had “no psychiatric disorders or neurologic diseases previously diagnosed, and no learning disabilities”?
Author Response
Responses to reviewers’ comments
All the responses and the modifications in the manuscript are written in blue.
Reviewer 1:
General Comments and Overall Evaluation (with limitations and strengths): Turning first to the general comments, this study is relevant to the mission of this journal and presents original data. The authors evaluated the autonomic response to pleasant affective touch in children with autism (n =20) and typically developing peers (n =30) matched by age. Three autonomic indices (pupil dilation, skin conductance, and heart rate) were recorded, and identical stimulations at two locations (i.e., CT high- and low-density) were compared. Overall, ASD children exhibited reduced autonomic responses and different ANS baseline values compared to TD children. Results also showed that the atypical autonomic responses to pleasant touch in children with autism were not specific to CT-fibers stimulation. However, some issues mitigate the impact of this manuscript (see below for more details), and there are several considerations that I believe need to be addressed before the study can be considered for publication.
Comments:
- Page 5, lines 218 – 221: the third objective of this study related to the sensory profiles measured by questionnaires needs a well-crafted theoretical background.
We have now added more detailed background in the Introduction to explain our third objective (p.4-5), and adjusted the Discussion accordingly (p.20).
- The authors measured heart rate, but why not Heart Rate Variability?
Heart rate variability (HRV) requires long time-windows to be properly estimated. Protocols calculating HRV parameters usually use 3 to 5 minutes windows (as we did in another paper, see Bufo et al. 2022). While some papers have started to use shorter time-windows (around 30 seconds) to evaluate HRV (Shaffer et al., 2014; Shaffer & Ginsberg, 2017), in our protocol we used stimulations lasting 4 seconds that were not appropriate for such an analysis.
- The sample size is quite small. Readers now expect to see a larger number of study participants, especially given the heterogeneity of symptoms in autism. Therefore, in my opinion, without a larger sample that allows robust statistical analyses, it will be difficult to make conclusions regarding this topic.
Our initial sample of ASD children was 37. We agree that, in the end, the number of ASD children for whom we could record and analyze the autonomic measures was small. However, when we compare the variability of our autonomic measures (assessed by the standard error values) between our 30 TD children, the TD children matched for each autonomic measure, and the ASD children for each autonomic measure, we found no difference. The standard error values are in the same range for all the groups. With this variability, we could observed reliable statistical differences in some conditions and groups (as assessed by our effect size η2p, going from 0.1 to 0.38). Of course, the results presented in this paper would need to be replicated in another, if possible larger, sample, as stated in our Discussion about the limits of the study (p.21). However, the heterogeneity in our ASD sample did not translate into a larger physiological variability than in our TD children (and we found very few correlations between our physiological results and clinical data).
- The authors describe the developmental characteristics of the participants (e.g., nonverbal IQ…) in the supplementary material. However, it is crucial to detail the main characteristics in the Method (Participants section). Given the heterogeneity of symptoms in autism, studies with variable samples (although with the same diagnosis) lead to inconsistent findings. Thus, it is crucial to understand confound variables, and the subgroups within the spectrum need to be specified experimentally. For instance, is crucial to understand how many children have intellectual disabilities, the level of autism symptom severity level… Furthermore, in this study, only children with autism who accepted to sit on the armchair, tolerated the physiological captors, and stayed calm during the data acquisition were considered for the analyses, probably biasing the sample towards a particular sensory profile. Thus, this sample is not representative of this population. We can have the same diagnosis, but quantitative differences can be obvious.
We have now added Table 1 in the main manuscript stating the characteristics of the total TD and ASD groups and the group of ASD children included but qualified as ‘non-tested’ (i.e. not tested because they did not accept to sit in the armchair or to be equipped with the captors, or not included in the analysis if there was not enough or non-exploitable data). The corresponding statistical analyses testing the differences between the tested and non-tested ASD children are presented in Supplementary Materials, as well as the repartition of the SPP2 scores to illustrate the clinical heterogeneity.
We argued that our approach aimed at testing ASD populations frequently excluded from research studies as unable to understand complex consigns or to produce verbal responses. Our tested sample exhibits IQ scores ranges showing that, indeed, we managed to test some ASD children with low IQ. However, we cannot deny that, even with rather non-intrusive physiological measurements, the sample of ASD children we could not test was biased towards children with low IQ and sensory atypicalities (Supplementary Materials). We have now discussed this bias in the Discussion (p.20 and p.21).
Interestingly, as already stated in the response to the previous comment, we found very few correlations between our autonomic responses and clinical data. We can thus expect that the results described in the present study are generalizable to the overall ASD population.
- Diagnosing ASD can be difficult, since there is no medical test, like a blood test, to diagnose the disorder. Therefore, we should use and report the comprehensive assessment for diagnosis of ASD, including the use of validated diagnostic instruments. Indeed, the clinical diagnosis is most valid and reliable when applying standardized diagnostic tools such as ADOS-2 and ADI-R. Thus, I suggest that the authors present more details about the diagnosis (mentioning the DSM-5 and CIM-10 criteria is not enough).
We have now extended our description of the diagnosis procedure in the University Hospital of Tours (p.5-6).
- The Children Sensory Profile 2 should be described in the Material (including their psychometric characteristics)
We have added a description of the Short Sensory Profile 2 in the Material (p.7) as well as in the Figure S1 and its legend.
- Page 5, line 228: How the authors verified that the TD children had “no psychiatric disorders or neurologic diseases previously diagnosed, and no learning disabilities”?
As now stated in the Material (p.5), we interviewed the parents and checked the medical booklet of the children.
Reviewer 2 Report
This was a very interesting report of the evaluation the autonomic response to pleasant affective touch (gentle stroking) in children with Autism Spectrum Disorders (ASD) and age-matched typically-developing (TD) peers which was assessed by measuring multiple Autonomic Nervous System (ANS) parameters, such as pupil diameter, skin conductance, and heart rate. I thought the applying of a specific methodology, combining a contrasting CT (C-tactile 13 fibers) high- vs low-density territories’ stimulations of two skin territories (CT high- and low-density, respectively forearm and palm of the hand) with ANS parameters, was particularly interesting. 50 6-12-year-old children took part in the study, among them 30 TD and 20 ASD children.
The authors found that ASD children exhibited reduced autonomic responses in comparison with their TD peers. The authors interpret the results obtained in terms of basal autonomic dysregulation and lower tactile autonomic evoked responses in ASD, assuming that it reflects lower arousal and relates to social disengagement in ASD children.
The manuscript as it currently stands is quite well structured and findings are clearly described. However, I’d suggest to the authors to add practical and theoretical relevance into the introduction part to highlight a research gap and to show its significance to the field.
Also the limitation section is necessary due to the fact that the sample you could access was unequal in gender composition and a fairly extensive age range was taken from 6 to 12 years.

Author Response
Responses to reviewers’ comments
All the responses and the modifications in the manuscript are written in blue.
Reviewer 2:
This was a very interesting report of the evaluation the autonomic response to pleasant affective touch (gentle stroking) in children with Autism Spectrum Disorders (ASD) and age-matched typically-developing (TD) peers which was assessed by measuring multiple Autonomic Nervous System (ANS) parameters, such as pupil diameter, skin conductance, and heart rate. I thought the applying of a specific methodology, combining a contrasting CT (C-tactile 13 fibers) high- vs low-density territories’ stimulations of two skin territories (CT high- and low-density, respectively forearm and palm of the hand) with ANS parameters, was particularly interesting. 50 6-12-year-old children took part in the study, among them 30 TD and 20 ASD children.
The authors found that ASD children exhibited reduced autonomic responses in comparison with their TD peers. The authors interpret the results obtained in terms of basal autonomic dysregulation and lower tactile autonomic evoked responses in ASD, assuming that it reflects lower arousal and relates to social disengagement in ASD children.
The manuscript as it currently stands is quite well structured and findings are clearly described. However, I’d suggest to the authors to add practical and theoretical relevance into the introduction part to highlight a research gap and to show its significance to the field.
We have added several parts in the Introduction to underline the clinical relevance of this research (p.2, p.5). We have also extended our background on the possible link between sensory questionnaires and sensory neurophysiological measures (p.4-5).
Also the limitation section is necessary due to the fact that the sample you could access was unequal in gender composition and a fairly extensive age range was taken from 6 to 12 years.
We have now extended the Discussion to mention specifically these limits (p.21).
Round 2
Reviewer 1 Report
The revised manuscript is much improved.